# COVID-19 Vaccination Did Not Increase the Risk of Potentially Related Serious Adverse Events: 18-Month Cohort Study in an Italian Province

**DOI:** 10.3390/vaccines11010031

**Published:** 2022-12-23

**Authors:** Maria Elena Flacco, Cecilia Acuti Martellucci, Graziella Soldato, Giuseppe Di Martino, Roberto Carota, Marco De Benedictis, Graziano Di Marco, Giustino Parruti, Rossano Di Luzio, Antonio Caponetti, Lamberto Manzoli

**Affiliations:** 1Department of Environmental and Prevention Sciences, University of Ferrara, 44121 Ferrara, Italy; 2Local Health Unit of Pescara, 65124 Pescara, Italy; 3Department of Medical and Surgical Sciences, University of Bologna, 40100 Bologna, Italy

**Keywords:** SARS-CoV-2, COVID-19, adverse events, cohort study, Italy

## Abstract

This cohort study on the entire population of an Italian Province assessed the incidence of potentially vaccine-related serious adverse events (PVR-SAEs) by COVID-19 vaccination status. From January 2021 to July 2022, we extracted all deaths and hospitalizations due to several cardiovascular diseases, pulmonary embolism, and deep vein thrombosis from National Healthcare System official data. During the follow-up, 5743 individuals died, and 2097 were hospitalized for PVR-SAEs. Vaccinated subjects (n = 259,821) did not show an increased risk of all-cause death, non-COVID death, or any PVR-SAEs, as compared to the unvaccinated (n = 56,494). These results were consistent across genders, age-classes, vaccine types, and SARS-CoV-2 infection status and did not vary in Cox models adjusting for age, gender, SARS-CoV-2 infection, and selected comorbidities. In the infected population, any dose of vaccine was associated with a lower likelihood of death and PVR-SAE. In the uninfected population, subjects who received one or two doses showed a significantly higher incidence of most outcomes, likely due to a large selection bias introduced by the Italian restriction policies targeting uninfected subjects who received less than three doses. In conclusion, COVID-19 vaccination was not associated with an increase of mortality or selected PVR-SAEs incidence. Further research is warranted to evaluate the long-term safety of COVID-19 vaccines.

## 1. Introduction

Like any other new preventive intervention, COVID-19 vaccines require in-depth population-based effectiveness and safety evaluations [1]. A number of field studies have confirmed the effectiveness of most circulating vaccines to control SARS-CoV-2 lethality [2,3,4,5,6], but very few controlled cohort studies assessing vaccine safety are publicly available [7,8,9,10,11,12]. In this cohort study, the entire population of an Italian province was followed up to 18 months to evaluate whether the incidence of selected potentially vaccine-related serious adverse events (PVR-SAEs) differed between unvaccinated and vaccinated individuals.

In Italy, during the vaccination campaign, among the pandemic control policies, the following regulations were imposed in a number of environments (including hospitals, universities, sport facilities, public transportation and other public places) and for all citizens ≥50 years of age: one COVID-19 vaccine dose for subjects who were previously infected with SARS-CoV-2, or three vaccine doses for uninfected subjects [13]. As a consequence, two situations emerged. On one side, the infected population opted to receive one, two, or three doses according to their personal beliefs. On the other side, among the uninfected, it is likely that the pressure to receive at least three doses introduced a selection bias, according to which individuals who received three doses may have been different from those who received only one or two doses, as the latter group was discouraged from receiving further vaccinations because of the occurrence of a disease or because they died before receiving the third dose. In order to account for the potential impact of this bias, we stratified the analyses by doses of vaccine and infection status.

## 2. Materials and Methods

This retrospective cohort analysis follows and expands two cohort studies which were focused on vaccine effectiveness [2,3]. Here, we included all individuals residing or domiciled in the province of Pescara, Italy, on 1 January 2021, aged six years or more, in order to compare the overall mortality and incidence of PVR-SAEs among subjects who received at least one dose of a COVID-19 vaccine versus the unvaccinated.

In the context of the Italian National Immunization Campaign, from 2 January 2021, the Pfizer–Biontech BNT162b2, Oxford–AstraZeneca ChAdOx1 nCoV-19, Moderna mRNA-1273, Janssen JNJ-78436735, and (more recently) NVX-CoV2373 vaccines were progressively administered to the population [14]. The Italian National Health System (NHS) provides vaccination, SARS-CoV-2 testing, and all healthcare services to residents free of charge. During follow-up, tests were mandatory for all individuals with symptoms such as fever or acute respiratory illness, for individuals who had contact with an infected person, and for all persons returning from travel abroad. Unvaccinated, or partially vaccinated uninfected individuals were subject to stringent work and movement restrictions, and on 15 February 2022, vaccination became compulsory for all Italians aged 50 years or more [13].

The flowchart of the study participants is shown in Figure 1. The cohort in this study was organized into four groups as follows. From 2 January 2021 (day of the first administered vaccine dose) to 31 July 2022 (end of follow-up):individuals who received only one dose of BNT162b2, mRNA-1273, ChAdOx1 nCoV-19, or NVX-CoV2373 were included in the group “1 dose only”;individuals who received only one dose of the JNJ-78436735 vaccine or only two doses of BNT162b2, mRNA-1273, ChAdOx1 nCoV-19, or NVX-CoV2373 were included in the group “2 doses only”;persons who received three or more doses of BNT162b2, mRNA-1273, ChAdOx1 nCoV-19, or NVX-CoV2373, or two or more vaccine doses, if one of the administered vaccines was JNJ-78436735, were included in the group “≥3 doses”;all individuals who received one or more doses of any of the above COVID-19 vaccines, i.e., categories A–C as described above, were included in the group “≥1 dose”.

When comparing the various types of vaccines, the subjects who received two or three different vaccines were included in a category named “mixed vaccines”.

For all subjects, the follow-up ended on the day of the outcome, or on 31 July 2022 if no outcome occurred. The start of follow-up varied depending on the comparison:When comparing unvaccinated subjects with either those who received one or more doses of vaccine (primary analysis) or those that received one dose only, the follow-up started the day of the first (or single) dose of vaccine for vaccinated subjects and on 2 January 2021 for the unvaccinated.When comparing unvaccinated subjects with subjects who received only two doses of vaccine, the follow-up started the day of the second dose of vaccine for vaccinated subjects and on 14 January 2022 (first day of the second dose administration) for the unvaccinated.When comparing unvaccinated subjects with those who received three or more doses of vaccine, the follow-up started the day of the third dose of vaccine for vaccinated individuals and on 1 July 2021 (first day of third dose administration) for unvaccinated subjects. The sample varied for the latter two analyses, as subjects who died before the start of the follow-up were excluded.

Subjects were considered “infected” if, before or during the follow-up, SARS-CoV-2 was detected at least once based on RT-PCR-Reverse transcription polymerase chain reaction testing using nasopharyngeal swabs in accredited regional laboratories [15]. For subjects who had a PVR-SAE, only those who had a positive swab before the outcome were considered as infected; those who were infected after the outcome were considered as uninfected for that specific analysis. Thus, the number of infected subjects varied in the analyses of each outcome. 

### 2.1. Outcomes

We selected 14 outcomes based upon frequency and clinical severity from a priority list of potential adverse events of special interest produced by the Brighton Collaboration and Coalition for Epidemic Preparedness, Innovation Partnership, Safety Platform for Emergency Vaccines, in their secondary analysis of serious adverse events reported in phase 3 randomized clinical trials of the Pfizer and Moderna mRNA COVID-19 vaccines in adults [16]: all-cause death, myocardial infarction, acute heart failure, ischemic stroke, hemorrhagic stroke, aortic aneurysm, peripheral artery aneurysm, cardiac arrest, coronary artery dissection, deep vein thrombosis, pulmonary embolism, myocarditis, pericarditis, and any PVR-SAE (subjects with at least one of any of the above diseases). The outcomes were hospital admissions that occurred during follow-up with the following ICD-9-CM codes in any diagnostic field: 410–410.99 (myocardial infarction); 428.21, 428.31, 428.41 (acute heart failure); 433.01, 433.11, 433.21, 433.31, 433.81, 433.91, 434.01, 434.11, 434.91, or 436 (ischemic stroke); 430–432 (hemorrhagic stroke); 441.00–441.9 (aortic aneurysm); 417.1 or 442.0–9 (peripheral artery aneurysm); 427.5 (cardiac arrest); 414.11 or 414.12 (coronary artery dissection); 452 or 453 (deep vein thrombosis); 415.19 (pulmonary embolism); 391.2, 429.0, 422.0, 422.90 or 422.99 (myocarditis); 391.0, 420.0, 420.90, 420.91, 420.99, 423.1 or 423.2 (pericarditis). Any of the above indicate the outcome “Any PVR-SAE”. Clearly, the total number of subjects identified as “Any PVR-SAE” was lower than the sum of individual PVR-SAEs, because some subjects had more than one event. The date of occurrence of the combined outcome “Any PVR-SAE” was the date of the first recorded outcome.

To account for some of the main potential confounders of the association between vaccination status and death or hospitalization [17], we used (a) the COVID-19 database, (b) the co-pay exemption database, and (c) administrative discharge abstracts from the last ten years to extract the following conditions for each resident: diabetes (ICD-9 cm codes in any diagnosis field—250.xx); hypertension (401.xx–405.xx); major cardiovascular or cerebrovascular diseases (410.xx–412.xx; 414.xx–415.xx; 428.xx or 433.xx–436.xx); chronic obstructive pulmonary diseases—COPD (491.xx–493.xx); kidney diseases (580.xx–589.xx); and cancer (140.xx–172.xx or 174.xx–208.xx). Additionally, we extracted all instances of PVR-SAEs which occurred before the start of the follow-up. In case of repeated admissions for the same condition, the date of the first episode was used.

### 2.2. Data Collection

All data regarding the administered vaccines, lab testing, and demographic, anamnestic, and clinical information are routinely collected and entered into the official datasets of the Italian NHS on a daily basis and are sent to the Italian Institute of Health [18]. All information were extracted from the official COVID-19, demographic, vaccination, hospital (Italian SDO), and co-pay exemption (“Esenzioni Ticket” file) datasets of the Pescara Local Health Unit, Italy. Encrypted fiscal codes were used to merge the information from the various datasets.

### 2.3. Statistical Analyses

The primary analyses compared the risk of each outcome of unvaccinated individuals versus subjects who received ≥1 vaccine dose. Secondary analyses compared the risk of each outcome between the unvaccinated and subjects who received a single dose, two doses only, or three or more doses. The univariate analyses were stratified by age class (0–29 y, 30–59 y, 60 + y), infection status (either never infected or infected before or after the date of the first vaccination), gender, and type of vaccine. The age classes were selected to be consistent with Italian Institute of Health reports [19] and with the policies of the Italian Government [20], which identified individuals aged 60 or more years as priority targets for vaccination.

Cox proportional hazards analysis was used to calculate the adjusted hazard ratios (HRs) of each outcome by vaccination status [21]. All the recorded potential confounders (age, gender, infections status, diabetes, hypertension, CVD, COPD, kidney diseases, cancer, and past hospitalization for the same outcome) were included a priori into all models. A minimum events-to-variable ratio of 10 was maintained in all multivariable models, and Schoenfeld’s test was performed to check the validity of proportional hazard assumptions [22]. Kaplan-Meier survival analysis was also used to examine the association between vaccination status and each outcome. The validity of constant incidence ratios up to follow-up was checked using Nelson-Aalen cumulative hazard estimates. A *p*-value < 0.05 was considered significant for all analyses, which were performed using Stata, version 13.1 (Stata Corporation, College Station, TX, USA, 2014).

## 3. Results

After the exclusion of 13,751 subjects aged five years or less, 67,431 hospitalizations of non-residents or domiciled individuals, and 164 erroneous fiscal codes, all of the 316,315 residents or domiciled individuals in the province of Pescara, Italy, were included in the analysis and followed from the start of the vaccination campaign (2 January 2021) up to 31 July 2022. The demographic characteristics, vaccine types and doses, and the proportion of selected risk factors and comorbidities, past episodes of PVR-SAEs, and SARS-CoV-2 infections are reported in Table 1.

Overall, 56,494 subjects were unvaccinated (17.9% of the population), 15,832 (5.0%) received only one vaccine dose, 51,684 (16.3%) received two doses, and 192,305 (60.8%) received three (n = 184,092) or four (n = 8213) doses. 

The mean age, as well as the prevalence of most comorbidities and PVR-SAEs, differed substantially across groups: those who received three or more doses were 12.5 years older, on average, than the unvaccinated and showed a higher prevalence of hypertension (16.9% vs. 7.5%, respectively), diabetes (6.4% vs. 3.1%), CVD (7.7% vs. 4.3%), COPD (3.7% vs. 2.6%), kidney diseases (3.7% vs. 1.0%), cancer (6.0% vs. 3.2%), and past episodes of PVR-SAEs (4.5% vs. 2.6%; all *p* < 0.05). In contrast, those who received at least three doses showed the lowest rate of SARS-CoV-2 infections (24.8% vs. more than 33% in all other groups).

Almost half of the population received only BNT162b2 (45.4%); 16.0% received only mRNA-1273; 1.2% received only the other vaccines, and 37.4% received mixed vaccines. The average follow-up was 428 ± 111 days; 561 ± 90 days for the unvaccinated and 399 ± 93 days for those who received at least one vaccine dose.

### 3.1. Risk of Death

During the follow-up, 5743 subjects died from any cause (1.82% of the population; Table 2). Infected subjects were younger on average than the uninfected (42.9 y vs. 50.7 y, respectively) and showed a lower death rate (1.11% vs. 2.18%; Appendix A).

The overall all-cause mortality was significantly higher among unvaccinated individuals than those who received at least one vaccine dose (4.23% vs. 1.29%, respectively; Table 2). The average monthly rate of deaths was 2.26 × 1000 individuals among the unvaccinated and 0.97 × 1000 among the vaccinated (with at least one dose). The lower incidence among the vaccinated was also apparent from the shape variation of the Kaplan-Meier estimates of time to death (Figure 2). The overall mortality was similar in all vaccine groups in the infected sample (ranging from 0.39% to 0.53%), while it varied widely in the uninfected group (Appendix A). In the latter population, mortality was substantially lower among subjects who received three or more doses (0.70%) than in the groups who received only one (5.70%) or two doses (6.65%), possibly reflecting the selection in the latter groups of subjects who died before they could be vaccinated further. This seems to be confirmed by the substantially higher proportion of early deaths in the partially vaccinated groups, as shown in Figure 3b (bottom), compared to unvaccinated individuals (Figure 3a—top).

When the analyses were further stratified (Appendix A), no substantial differences were observed across genders, while in the older population, the absolute difference in mortality between vaccinated and unvaccinated individuals widened, with an overall death rate of 19.1% for the unvaccinated but 3.63% for those who received at least one dose. Regarding vaccine types, the lowest death rate was observed among subjects who received two different vaccines (0.41%; Appendix A).

Multivariable analyses largely confirmed the univariate results. Adjusting for age, gender, SARS-CoV-2 infection, hypertension, diabetes, CVD, COPD, kidney disease, cancer, and past PVR-SAE occurrence, the risk of all-cause death was significantly and substantially lower for individuals who received at least one vaccine dose, as compared with the unvaccinated (HR: 0.19; 95% CI: 0.18–0.20; Table 3). Significantly lower risks of death were also observed for all vaccinated groups among the infected and among uninfected subjects who received three or more doses (HR: 0.66; 0.64–0.69). Conversely, uninfected subjects who received only one or two doses showed a significantly higher mortality than the unvaccinated.

### 3.2. Potentially Vaccine-Related Serious Adverse Events (PVR-SAE)

During the follow-up, 2097 subjects had a new episode of PVR-SAE (0.66% of the population; Table 2). Compared to the unvaccinated, subjects who received at least one vaccine dose showed a significantly lower overall incidence of PVR-SAEs (0.97% vs. 0.60%, respectively; *p* < 0.001) and comparable average monthly rates (0.52 × 1000 vs. 0.45 × 1000, respectively; Table 2). A similar overall incidence of PVR-SAE was observed among the infected, while in the uninfected group, subjects who received only one or two doses showed a substantially higher proportion of events than those who received three or more doses (Appendix A).

In all groups, males showed a higher incidence of PVR-SAEs, the vast majority of which were recorded in the older group (Appendix A). As for all-cause mortality, the lowest incidence of PVR-SAE was observed among subjects who received two different vaccines (0.26%).

Compared with the unvaccinated, subjects who received at least one vaccine dose showed a significantly lower likelihood of PVR-SAE (adjusted HR: 0.39; 95% CI: 0.36–0.43; Table 3). Significantly lower risks of PVR-SAE were also observed for all vaccinated groups among the infected and among uninfected subjects who received three or more doses (HR: 0.82; 0.76–0.87). In contrast, uninfected individuals that received only one or two doses showed a significantly higher incidence of PVR-SAE than the unvaccinated.

Regarding individual PVR-SAEs, the most frequently observed during the follow-up were ischemic stroke (n = 495; 0.16% of the sample), cardiac arrest (n = 477), myocardial infarction (n = 475; 0.15%), and hemorrhagic stroke (n = 306; 0.10%), while the least common were coronary dissection (n = 6) and myocarditis (n = 9; Table 2). Stratified univariate analyses of all the selected PVR-SAEs are reported in Appendix A. In multivariable analyses, the likelihood of none of the individual PVR-SAEs was significantly higher among subjects who received at least one dose of vaccine, as compared with the unvaccinated (Table 3). There were no significant differences between vaccinated (with at least one dose) and unvaccinated subjects in the risk of myocardial infarction, aortic aneurysm, or myocarditis/pericarditis, while the likelihood of ischemic stroke, hemorrhagic stroke, cardiac arrest, deep vein thrombosis, and pulmonary embolism was significantly lower among the vaccinated (all *p* < 0.001; Table 3). The same trend that was observed for the total PVR-SAEs and death emerged for most individual PVR-SAEs: among infected subjects, all vaccinated groups showed a lower risk than the unvaccinated; among uninfected individuals, those who received three or more vaccine doses showed a significantly lower risk, while the groups who received only one or two vaccine doses showed a significantly higher incidence.

## 4. Discussion

In the entire population of an Italian province, followed for an average of 14 months, subjects who received one or more doses of COVID-19 vaccines did not show an increased risk of death from any cause, death unrelated to SARS-CoV-2 infection, or any of the recorded potentially vaccine-related serious adverse events requiring hospitalization. Furthermore, some of the outcomes (all-cause mortality, and incidence of stroke, cardiac arrest, thrombosis, and pulmonary embolism) were significantly less frequent among the vaccinated. These findings were consistent across genders, age-classes, most frequently administered vaccine types, and SARS-CoV-2 infection status and remained so after adjusting for previous episodes of disease and several potential confounders. Importantly, strikingly different results were observed in the uninfected population, between subjects who received three or more doses and those who received only one or two doses. The latter groups showed a significantly higher incidence of most outcomes, probably due to the selection bias introduced by the stringent Italian restriction policies concerning subjects who received less than three doses [13].

While the effectiveness of these vaccines against severe COVID-19 has been documented in several population studies [2,3,4,5,6], some primary studies [7,9,10,11,23] and secondary analyses of serious adverse events reported (a) in the randomized clinical trials of Pfizer and Moderna mRNA COVID-19 vaccines in adults [16,24], (b) in the U.S.A. Vaccine Adverse Event Reporting System (VAERS) passive surveillance data [25,26], and (c) in the National registries of Denmark, Finland, and Norway [27], have raised concerns about the safety or risk-benefit profile of COVID-19 vaccination, especially in population subsets with the lowest infection-fatality rate [28]. However, other studies did not show a higher risk of serious adverse events in mRNA vaccine recipients [29,30,31]. Most importantly, to date, the only two population-based primary studies with a control group available found significantly lower non-COVID mortality among vaccinated individuals after 28 days [12] and 7.5 months of follow-up [8].

Both epidemiological and immunological studies showed that COVID-19 vaccination, received either before or after SARS-CoV-2 infection (known as “hybrid immunity”), is able to increase the protection against the virus as compared with natural immunity [15,32]. No data are available, however, on the potential impact on vaccination safety of having received the vaccine prior or after infection with SARS-CoV-2. We thus further stratified the infected population according to the timing of the infection (Appendix A), finding no substantial differences in the overall results between the individuals who were infected before or after vaccination.

As mentioned, while the results were homogeneous for the infected population, the incidence of several outcomes, including death and total PVR-SAEs, was substantially higher among uninfected subjects who received one or two vaccine doses, as compared to those who received three of more doses. This counterintuitive finding may be explained, at least in part, by selection bias. Indeed, during the pandemic, millions of Italians were obliged to receive three vaccine doses to retain their job or access a number of environments (including universities, hospitals, and public places) [13]. However, subjects who were infected with SARS-CoV-2 were granted a six-month extension, after which they were mandated to receive only one vaccine dose. As a result, two scenarios emerged. On one side, among uninfected subjects, given the strong pressure to receive three doses, there were only two reasonable categories: those who refused the vaccine in toto and those who wanted to receive three doses, as receiving one or two doses only would not have met the official requirements. This is indeed apparent in the sample: 83.2% of vaccinated uninfected subjects received at least three doses. It may thus be assumed that many of those who received only one or two doses were discouraged from receiving further vaccinations because of the occurrence of a disease, or that they died before receiving the third dose. Thus, these subjects were selectively included in the groups “one dose only” and “two doses only”. In contrast, the infected population did not experience the above selection bias: the vaccination mandate for them was far less stringent, and individuals opted to receive one, two, of three doses according to their personal convictions, which gave rise to very homogeneous results across all vaccine groups. In any case, and most importantly, even among uninfected individuals, the overall comparison, which included all subjects who received at least one dose, showed no increase in the incidence of any outcome among vaccinated subjects, as compared with the unvaccinated.

### Strengths and Limitations

To date, on this topic, this is the controlled cohort study with the longest follow-up and the first to adjust the analyses for multiple comorbidities and infection status, which likely reduced the misclassification bias related to defining COVID-19 and non-COVID-19 deaths based solely on a temporal criterion [28]. Additionally, the availability of multiple, official databases may have permitted to capture the vast majority of serious events.

However, the study has limitations that must be considered. First, no inferences can be derived on the vaccination-related risk of mild or moderate adverse events, as well as for longer follow-ups. Second, consistently with the observational nature of the study, there may be confounding due to the healthy vaccinee effect, wherein vaccinated people may be expected to adopt less risky behaviors [8]. Third, we based our definition of infection on the available lab data, which, in turn, was based on positive swabs; however, the rates of infection are certainly underestimated, as the existing monitoring system cannot detect all asymptomatic infections [33,34]. Although this issue may severely alter the estimates of the SARS-CoV-2 incidence-fatality rate [35], its impact on comparisons between vaccinated and unvaccinated individuals remains to be ascertained, as it is not known whether the proportion of undetected infections differs by vaccination status [36]. Finally, and most importantly, the overall results among the infected and uninfected subjects were similar. 

## 5. Conclusions

In the entire population of an Italian province, followed for an average of 14 months, individuals who received one or more doses of COVID-19 vaccines did not show an increased risk of death for any cause, death unrelated to SARS-CoV-2 infection, or any of the selected potentially vaccine-related serious adverse events requiring hospitalization (myocardial infarction, acute heart failure, cardiac arrest, ischemic or hemorrhagic stroke, coronary artery dissection, aortic or peripheral aneurysm, pulmonary embolism, deep vein thrombosis, and myocarditis, or pericarditis). These findings were consistent across genders, age-classes, most frequently administered vaccine types, and SARS-CoV-2 infection status and remained so after adjustment for previous episodes of disease and several potential confounders. Further research in the coming years will be required to evaluate the long-term safety of COVID-19 vaccines.

## Figures and Tables

**Figure 1 vaccines-11-00031-f001:**
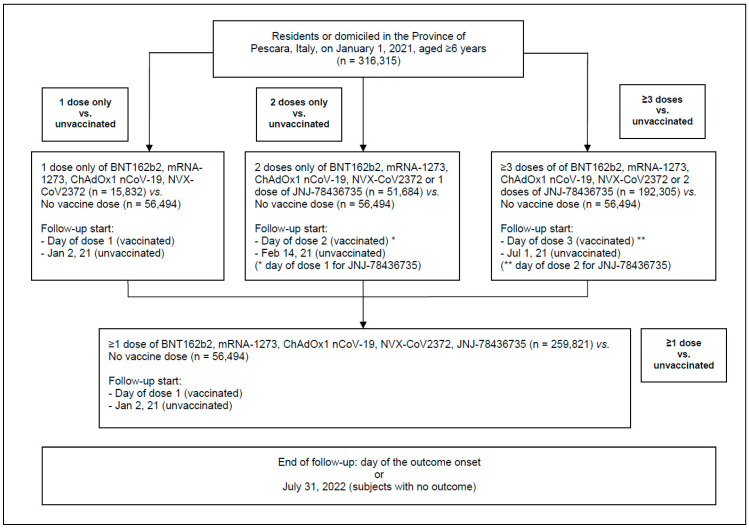
Flowchart of study participants.

**Figure 2 vaccines-11-00031-f002:**
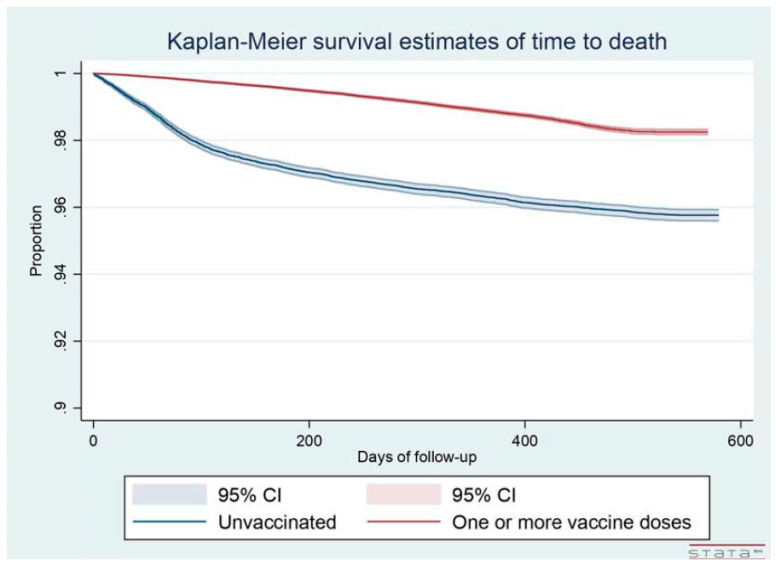
Kaplan-Meier estimates of time to death by vaccination status (at least one dose of COVID-19 vaccines vs. unvaccinated).

**Figure 3 vaccines-11-00031-f003:**
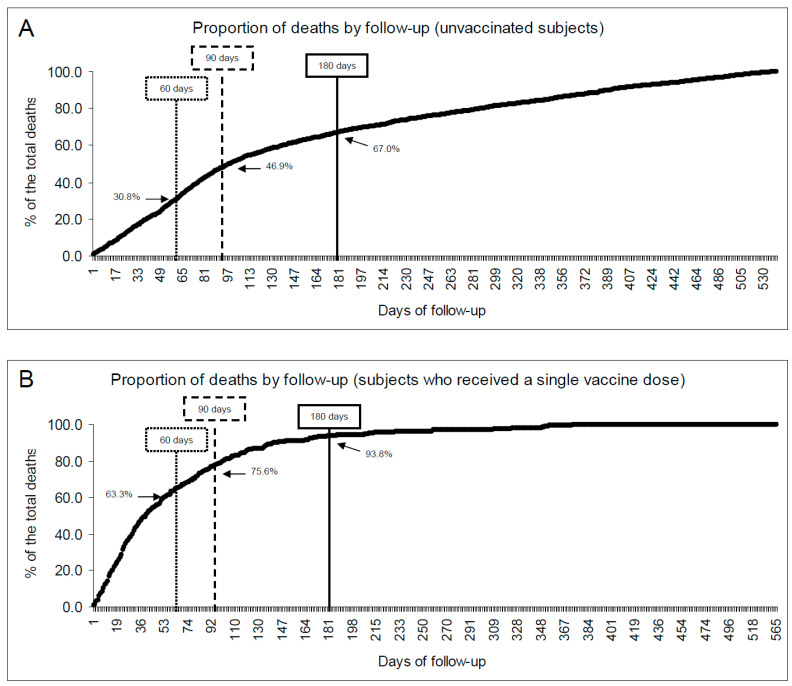
Cumulative proportion of the deaths by follow-up time. Figure 3A (**top**): among unvaccinated subjects; Figure 3B (**bottom**) among partially vaccinated subjects (those who received a single COVID-19 vaccine dose).

**Table 1 vaccines-11-00031-t001:** Main characteristics of the sample, overall and by COVID-19 vaccine status.

	Total Sample	Unvaccinated	1 Dose ^I^	2 Doses ^II^	≥3 doses ^III^	≥1 Dose ^IV^
	(n = 316,315)	(n = 56,494)	(n = 15,832)	(n = 51,684)	(n = 192,305)	(n = 259,821)
Male gender, %	48.9	50.5	51.0	50.3	47.9	48.5
Mean age in years (SD)	48.0 (22.0)	40.8 (22.8)	41.8 (20.6)	38.1 (21.9)	53.3 (20.2)	49.6 (21.5)
*Age category in years, %*						
6–29	24.0	33.6	32.7	41.1	16.0	22.0
30–59	44.8	46.2	47.8	42.5	44.7	44.4
60 or more	31.2	20.2	19.5	16.4	39.4	33.6
*Risk factors and comorbidities* ^V^ *, %*						
Hypertension	13.2	7.5	8.6	7.4	16.9	14.5
Diabetes	5.1	3.1	3.2	3.0	6.4	5.5
CVD	6.3	4.3	4.0	4.1	7.7	6.8
COPD	3.3	2.6	2.4	3.2	3.7	3.5
Kidney disease	1.3	1.0	0.5	1.0	1.5	1.3
Cancer	4.8	3.2	3.0	2.7	6.0	5.1
Past potentially vaccine-related serious adverse events (PVR-SAE) ^VI^, %						
PVR-SAE, total	3.8	2.6	2.9	2.4	4.5	4.0
Myocardial infarction	1.4	0.8	1.0	0.7	1.7	1.5
Acute heart failure	0.06	0.03	0.03	0.04	0.07	0.06
Cardiac arrest	0.03	0.04	0.04	0.02	0.04	0.03
Ischemic stroke	1.1	0.9	0.9	0.7	1.3	1.2
Hemorrhagic stroke	0.4	0.3	0.3	0.3	0.5	0.4
Coronary artery dissection	0.01	0.00	0.00	0.00	0.01	0.01
Aortic aneurysm	0.5	0.3	0.3	0.3	0.7	0.6
Peripheral aneurysm	0.10	0.09	0.06	0.08	0.12	0.11
Pulmonary embolism	0.2	0.2	0.2	0.1	0.2	0.2
Deep vein thrombosis	0.2	0.2	0.2	0.1	0.2	0.2
Myocarditis	0.04	0.05	0.04	0.04	0.05	0.04
Pericarditis	0.2	0.1	0.2	0.2	0.3	0.2
*Type of vaccine, %*						
BNT162b2	45.4	--	51.2	66.6	38.4	45.4
mRNA-1273	16.0	--	47.4	15.5	11.9	16.0
ChAdOx1 nCoV-19	0.9	--	1.2	13.0	0.0	0.9
JNJ-78436735	0.2	--	--	2.1	0.0	0.2
NVX-CoV2373	0.1	--	0.2	0.1	0.0	0.1
Mixed ^VII^	37.4	--	--	2.8	49.7	37.4
Infected with SARS-CoV-2 ^VIII^, %	34.3	40.0	74.6	51.2	24.8	33.1
Mean follow-up, days (SD)	428 (111)	561 (90) ^IX^	204 (70)	279 (91)	209 (37)	399 (93) ^X^

^I^ Subjects who received only one dose of BNT162b2, mRNA-1273, ChAdOx1 nCoV-19, or NVX-CoV2373. ^II^ Subjects who received only one dose of JNJ-78436735 or only two doses of the BNT162b2, mRNA-1273, ChAdOx1 nCoV-19, or NVX-CoV2373 vaccines. ^III^ Subjects who received three or more doses of one of the BNT162b2, mRNA-1273, ChAdOx1 nCoV-19, or NVX-CoV2373 vaccines, or two or more doses of these vaccines if one of the administered doses was JNJ-78436735. ^IV^ Subjects who received one or more doses of COVID-19 vaccines between 2 January 2021 and 31 July 2022 (end of follow-up). ^V^ Subjects with selected comorbidities (please see the text for details). ^VI^ Potentially Vaccine-related serious adverse events (please see the text for details). The total number of PVR-SAEs does not equal the sum of individual PVR-SAEs, because some subjects had more than one event. ^VII^ Subjects who received two or three different vaccines. ^VIII^ Subjects who had at least one positive SARS-CoV-2 swab before the outcome. ^IX^ For unvaccinated individuals, the follow-up started (a) on 2 January 2021 for the comparison between unvaccinated and the groups “≥1 Doses” and “1 Dose” (mean follow-up shown in the table); (b) on 14 January 2021 for the comparison between unvaccinated and the group “2 Doses” (mean follow-up 547 days); (c) on 1 July 2021 for the comparison between the unvaccinated and the group “≥3 Doses” (mean follow-up 337 days). ^X^ The follow-up started (a) after the first dose for all subjects in the groups “≥1 Dose” and “1 Dose”; (b) after the second dose (or the first of JNJ-78436735 vaccine) for the group “2 Doses”; (c) after the third dose (or the second dose if a JNJ-78436735 vaccine was administered) for the group “≥3 Doses”. Thus, a subject who received two doses had different follow-ups: a longer one (starting from the first dose) in the comparison between unvaccinated and all subjects who received at least one dose; and a shorter one (starting from the second dose) in the comparison between unvaccinated and the group who received only two doses.

**Table 2 vaccines-11-00031-t002:** Main outcomes * by COVID-19 vaccine status. (A) All-cause death and non-COVID-19 death by vaccine status. (B) Potentially vaccine-related serious adverse events (PVR-SAEs) by vaccine status.

(A)	Total Sample	Unvaccinated	1 Dose ^I^	2 Doses ^II^	≥3 Doses ^III^	≥1 Dose ^IV^
	(N = 316,315)	(N = 56,494)				(N = 259,821)
Death, overall (N)			(15,832)	(51,684)	(192,305)	
% (n)	1.82 (5743)	4.23 (2392)	1.74 (275)	3.52 (1819)	0.65 (1257)	1.29 (3351)
Mean monthly rate x1000	1.27	2.26	2.55	3.78	0.94	0.97
Non-COVID-19 Death						
% (n)	1.54 (4873)	3.21 (1815)	1.62 (256)	3.32 (1715)	0.57 (1087)	1.18 (3058)
Mean monthly rate x1000	1.08	1.72	2.38	3.57	0.81	0.88
Death among the uninfected only (N)	(207,721)	(33,899)	(4015)	(25,243)	(144,564)	(173,822)
% (n)	2.18 (4536)	4.75 (1611)	5.70 (229)	6.65 (1679)	0.70 (1017)	1.68 (2925)
Mean monthly rate x1000	1.51	2.55	7.34	7.95	1.01	1.23
**(B)**	**Total Sample**	**Unvaccinated**	**1 Dose ^I^**	**2 Doses ^II^**	**≥3 Doses ^III^**	**≥1 Dose ^IV^**
	**(N = 316,315)**	**(N = 56,494)**				**(N = 259,821)**
PVR-SAE, total * (N)	(N = 316,315)	(N = 56,494)	(15,913)	(52,067)	(191,841)	(N = 259,821)
% (n)	0.66 (2097)	0.97 (549)	0.88 (140)	1.57 (818)	0.31 (590)	0.60 (1548)
Mean monthly rate x1000	0.46	0.52	1.29	1.69	0.44	0.45
Myocardial infarction (N)			(15,862)	(51,816)	(192,143)	
% (n)	0.15 (475)	0.12 (67)	0.25 (39)	0.38 (198)	0.09 (171)	0.16 (408)
Acute heart failure (N)			(15,832)	(51,688)	(192,301)	
% (n)	0.01 (38)	0.01 (4)	0.01 (1)	0.03 (18)	0.01 (15)	0.01 (34)
Cardiac arrest (N)			(15,832)	(51,688)	(192,301)	
% (n)	0.15 (477)	0.39 (221)	0.16 (26)	0.27 (141)	0.05 (89)	0.10 (256)
Ischemic stroke (N)			(15,855)	(51,790)	(192,176)	
% (n)	0.16 (495)	0.18 (103)	0.22 (35)	0.44 (229)	0.07 (128)	0.15 (392)
Hemorrhagic stroke (N)			(15,840)	(51,733)	(192,248)	
% (n)	0.10 (306)	0.13 (75)	0.11 (18)	0.23 (120)	0.05 (93)	0.09 (231)
Coronary dissection (N)			(15,832)	(51,685)	(192,304)	
% (n)	0.00 (6)	0.00 (2)	0.00 (0)	0.00 (2)	0.00 (2)	0.00 (4)
Aortic aneurism (N)			(15,846)	(51,741)	(192,234)	
% (n)	0.06 (192)	0.04 (21)	0.11 (18)	0.16 (81)	0.04 (72)	0.07 (171)
Peripheral aneurism (N)			(15,835)	(51,697)	(192,289)	
% (n)	0.01 (47)	0.01 (8)	0.02 (3)	0.04 (20)	0.01 (16)	0.02 (39)
Pulmonary embolism (N)			(15,835)	(51,711)	(192,275)	
% (n)	0.05 (162)	0.10 (56)	0.06 (9)	0.12 (63)	0.02 (34)	0.04 (106)
Deep vein thrombosis (N)			(18,539)	(51,697)	(192,285)	
% (n)	0.03 (101)	0.05 (30)	0.06 (9)	0.07 (38)	0.01 (24)	0.03 (71)
Myocarditis (N)			(15,832)	(51,690)	(192,299)	
% (n)	0.00 (9)	0.00 (1)	0.00 (0)	0.01 (7)	0.00 (1)	0.00 (8)
Pericarditis (N)			(15,836)	(51,693)	(192,292)	
% (n)	0.01 (37)	0.01 (6)	0.03 (5)	0.04 (19)	0.00 (7)	0.01 (31)

* Potentially vaccine-related serious adverse events. At least one hospital admission for the selected diseases from the start of follow-up (2 January 2021 for the unvaccinated; the day of the first dose for the groups “≥1 Dose” and “1 Dose”; the day of the second dose for the group “2 Doses”; the day of the third dose for the group “≥3 Doses”) up to 31 July 2022. Deaths were extracted from the official demographic database (Italian “Anagrafica”) and the hospital admission discharge abstracts (Italian “SDO”). The events were extracted from the Italian SDO database; please see the text for details. The total number of PVR-SAEs does not equal the sum of individual PVR-SAEs, because some subjects had more than one event. ^I^, ^II^, ^III^,^IV^ Please see the footnotes of Table 1 I, II, III, IV, respectively.

**Table 3 vaccines-11-00031-t003:** Adjusted hazards ratios (95% confidence interval—CI) ^I^ of death and potentially vaccine-related serious adverse events (PVR-SAEs) ^II^, overall and by infection status. The unvaccinated group is the reference category for all analyses.

	1 Dose ^III^	2 Doses ^IV^	≥3 Doses ^V^	≥1 Dose ^VI^
Outcomes	HR (95% CI)	HR (95% CI)	HR (95% CI)	HR (95% CI)
Death, total sample	0.82 (0.72–0.94)	1.10 (1.06–1.13)	0.66 (0.64–0.68)	0.19 (0.18–0.20)
Uninfected only ^VII^	2.08 (1.80–2.39)	1.29 (1.25–1.34)	0.66 (0.64–0.69)	0.22 (0.20–0.23)
Infected only ^VII^	0.19 (0.14–0.25)	0.55 (0.50–0.60)	0.66 (0.62–0.70)	0.11 (0.10–0.13)
All PVR-SAEs, total sample	1.59 (1.30–1.93)	1.43 (1.35–1.51)	0.76 (0.72–0.80)	0.39 (0.36–0.43)
Uninfected only ^VII^	3.69 (3.00–4.55)	1.74 (1.63–1.85)	0.82 (0.76–0.87)	0.52 (0.47–0.59)
Infected only ^VII^	0.23 (0.14–0.39)	0.54 (0.46–0.65)	0.58 (0.51–0.66)	0.12 (0.09–0.15)
Myocardial infarction	4.96 (3.20–7.67)	2.10 (1.81–2.44)	0.95 (0.82–1.09)	0.84 (0.64–1.09)
Cardiac arrest	0.49 (0.32–0.75)	0.95 (0.85–1.05)	0.64 (0.57–0.72)	0.18 (0.15–0.22)
Ischemic stroke	2.60 (1.74–3.88)	1.71 (1.52–1.92)	0.76 (0.67–0.86)	0.50 (0.40–0.63)
Hemorrhagic stroke	1.81 (1.06–3.10)	1.51 (1.31–1.76)	0.75 (0.65–0.87)	0.45 (0.34–0.59)
Aortic aneurysm	7.51 (3.92–14.4)	2.21 (1.73–2.82)	0.85 (0.69–1.04)	0.96 (0.61–1.52)
Pulmonary embolism	1.00 (0.48–2.10)	1.31 (1.09–1.57)	0.67 (0.55–0.82)	0.30 (0.22–0.42)
Deep vein thrombosis	2.78 (1.27–6.09)	1.45 (1.13–1.85)	0.71 (0.55–0.91)	0.37 (0.24–0.57)
Myocarditis/Pericarditis	5.86 (1.73–19.8)	2.10 (1.37–3.22)	0.67 (0.42–1.06)	0.74 (0.32–1.67)

^I^ Based on Cox proportional hazards models, adjusted for age, gender, infection status, diabetes, hypertension, cardiovascular or cerebrovascular disease, chronic obstructive pulmonary disease, kidney disease, cancer, and outcome occurrence before the start of follow-up. For all analyses, the reference category is the unvaccinated group. ^II^ At least one hospital admission for the selected diseases from the start of follow-up (2 January 2021 for the unvaccinated; the day of the first dose for the groups “≥1 Dose” and “1 Dose”; the day of the second dose for the group “2 Doses”; the day of the third dose for the group “≥3 Doses”) up to 31 July 2022. The outcomes “acute heart failure”, “peripheral aneurysm”, and “coronary artery dissection” were not included in the multivariable analyses due to the scarce number of events. For the same reason, myocarditis and pericarditis have been grouped. ^III^, ^IV^, ^V^, ^VI^ Please see the footnotes of Table 1 I, II, III, IV, respectively. ^VII^ Infected subjects had at least one positive SARS-CoV-2 swab before the outcome. The Cox models stratified by infection were not adjusted for infection status.

## Data Availability

The data presented in this study are available upon reasonable request from the corresponding author.

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
