# Peer review of "COVID-19 Vaccination Did Not Increase the Risk of Potentially Related Serious Adverse Events: 18-Month Cohort Study in an Italian Province"

_vaccines, 2022, doi:10.3390/vaccines11010031_

Round 1
Reviewer 1 Report
The manuscript by Flacco and co-workers describes the longest follow-up of COVID-19 vaccinees to demonstrated no increased risk of serious vaccine-related adverse effects or mortality compared to unvaccinated individuals. This was a comprehensive well-conducted retrospective cohort study conducted with a large number of subjects which of a great interest to both health work professionals and the community in general.
Few issues should be addressed in order to improve the manuscript:
1) The following sentence in the abstract “…selection bias introduced by the Italian restriction policies targeted 29 to the uninfected subjects who received less than three doses.”, is rather a vague statement, perhaps authors should be more specific. The actual restriction polices will be only mentioned at the discussion section. Therefore, authors should provide more detailed information within the Introduction section.
2) Authors divided the vaccinated population analyzed into four distinct groups; however it was not clear why authors included a "≥1 dose" group. How is it different from groups “one dose” and “two doses”, from the criteria description, people included in the previous one and two dose groups wouldn’t also be part on this "≥1 dose" group?
3) Authors should provide detailed legends for figures 1 and 2, since only a figure title is present. For comparison purposes figures 2 top and bottom?
Lane 260 while instead of why.
Author Response
I-1. The Reviewer wrote: "The manuscript by Flacco and co-workers describes the longest follow-up of COVID-19 vaccinees to demonstrate no increased risk of serious vaccine-related adverse effects or mortality compared to unvaccinated individuals. This was a comprehensive well-conducted retrospective cohort study conducted with a large number of subjects which of a great interest to both health work professionals and the community in general".
The Reviewer also wrote: "Few issues should be addressed in order to improve the manuscript: the following sentence in the abstract “…selection bias introduced by the Italian restriction policies targeted to the uninfected subjects who received less than three doses.”, is rather a vague statement, perhaps authors should be more specific. The actual restriction polices will be only mentioned at the discussion section. Therefore, authors should provide more detailed information within the Introduction section".
We agree and accordingly added the following paragraph in the Introduction section: "In Italy, during the vaccination campaign, among the pandemic control policies, the following obligations were set to access a number of environments (including universities, hospitals, sport facilities and any public places), and for all the citizens aged 50 years or more: one COVID-19 vaccine dose for the subjects who were previously infected with SARS-CoV-2, or three vaccine doses for the uninfected subjects [14]. As a result, two different scenarios emerged. On one side, the infected population opted to receive one, two of three doses according to their personal convincement. On the other side, among the uninfected, it is likely that the strong pressure to receive at least three doses introduced a selection bias, according to which the subjects who received three doses might be different from those who received only one or two doses, as the latter individuals were reasonably discouraged to further vaccinations because of the occurrence of a disease, or since they deceased before the third dose. In order to account for the potential impact of the above mentioned bias, we will stratify the analyses by doses of vaccine and infection status".
I-2. The Reviewer wrote: "Authors divided the vaccinated population analyzed into four distinct groups; however it was not clear why authors included a "≥1 dose" group. How is it different from groups “one dose” and “two doses”, from the criteria description, people included in the previous one and two dose groups wouldn’t also be part on this "≥1 dose" group?"
We agree that the distinction between "≥1 dose" group, "one dose" and "two or more doses" requires a clarification, and we apologize for not having clarified the issue before: please acknowledge that the group "≥1 dose" includes: (a) the subgroup of partially vaccinated individuals who received only one dose of BNT162b2, mRNA-1273, ChAdOx1 nCoV-19 or NVX-CoV2373 vaccines; (b) those subjects who received two doses of the above mentioned vaccines, or one dose of JNJ-78436735 vaccine (characterized by a single-dose primary schedule); (c) those subjects who received three or more doses of mRNA-1273, ChAdOx1 nCoV-19 or NVX-CoV2373 vaccines, or two or more doses of JNJ-78436735 vaccine (booster doses). In order to clarify this point, please acknowledge that the previous lines 62-74 in the methods:
"From 2 January 2021 (date of the administration of the first vaccine dose), up to July 31, 2022 (end of follow-up):
- the subjects who received only one dose of BNT162b2, mRNA-1273, ChAdOx1 nCoV-19 or NVX-CoV2373 vaccines, were included in the group “1 dose”;
- the subjects who received only one dose of JNJ-78436735 vaccine, or only two doses of BNT162b2, mRNA-1273, ChAdOx1 nCoV-19 or NVX-CoV2373 vaccines, were included in the group “2 doses”;
- the subjects who received three or more doses of either BNT162b2, mRNA-1273, ChAdOx1 nCoV-19 or NVX-CoV2373 vaccines, or two or more doses of vaccines, if one of the administered vaccines was JNJ-78436735, were included in the group “≥3 doses”;
- all the subjects who received one or more doses of any of the above COVID-19 vaccines were included in the group "≥1 dose"
were rephrased as follows:
"From 2 January 2021 (date of the administration of the first vaccine dose), up to July 31, 2022 (end of follow-up):
- the subjects who received only one dose of BNT162b2, mRNA-1273, ChAdOx1 nCoV-19 or NVX-CoV2373 vaccines, were included in the group “1 dose only”;
- the subjects who received only one dose of JNJ-78436735 vaccine, or only two doses of BNT162b2, mRNA-1273, ChAdOx1 nCoV-19 or NVX-CoV2373 vaccines, were included in the group “2 doses”;
- the subjects who received three or more doses of either BNT162b2, mRNA-1273, ChAdOx1 nCoV-19 or NVX-CoV2373 vaccines, or two or more doses of vaccines, if one of the administered vaccines was JNJ-78436735, were included in the group “≥3 doses”;
- all the subjects who received one or more doses of any of the above COVID-19 vaccines, listed in the above a-c categories, were included in the group "≥1 dose".
I-3. The Reviewer wrote: "Authors should provide detailed legends for figures 1 and 2, since only a figure title is present. For comparison purposes figures 2 top and bottom?"
We agree, and thank the Reviewer. Accordingly, the previous Figure 2 caption "Cumulative proportion of the deaths by follow-up time, among the unvaccinated subjects (top) and those who received a single COVID-19 vaccine dose (bottom)" was replaced with "Cumulative proportion of the deaths by follow-up time. Figure 2a (top): among unvaccinated subjects; Figure 2b (bottom): among partially vaccinated subjects (those who received a single COVID-19 vaccine dose)".
I-4. The Reviewer wrote: "Line 260 while instead of why".
We thank the Reviewer, and replaced "why" with "while" where appropriate. We apologize for the oversight.
Reviewer 2 Report
This cross-sectional study is a useful study in terms of long observation follow-up in groups administered different doses of vaccine. It also indicates different vaccine and dose vaccinations of the population during the covid period.
In fact, the aim of the study is to investigate the side effects of the vaccine in people who have had the disease or not. However, distinguishing side effects for different vaccines could provide more information.
1. Line 207- 208, ” Is the number of vaccines administered to these people less than other groups? It would be appropriate to clarify this issue. Evaluate according to the sentence below.
“During the follow-up, 5743 subjects deceased for any cause (1.82% of the population;207
Table 2). The infected subjects were younger on average than the uninfected (42.9y vs.
50.7y, respectively) 208and showed a lower death rate (1.11% vs. 2.18%; Table S1.
2. Line 236. ; -. It would be appropriate to examine the cause of mortality in these cases; these are the underlying malignancy etc. Is it people who have a problem who can't develop enough protective antibodies?
“Conversely, the uninfected subjects who received only one or two doses showed a significantly higher mortality than the unvaccinated”
3. Lİne 243-245; -Why did those who received two doses of the vaccine have more side effects than those who received three or more doses? Is there a situation like immune blinding here?
“A similar overall incidence of PVR-SAE was observed among the infected, while in the uninfected group the subjects who received only one or two doses showed a substantially higher proportion of events than those who received three or more doses. “
4. Line 251-252; Please comment on the following sentence in the discussion section.
“Compared with the unvaccinated, the subjects who received at least one vaccine
dose showed a significantly lower likelihood of PVR-SAE (adjusted HR: 0.39; 95% CI:
0.36-0.43; Table 3)”.
Author Response
II-1. The Reviewer wrote: "This cross-sectional study is a useful study in terms of long observation follow-up in groups administered different doses of vaccine. It also indicates different vaccine and dose vaccinations of the population during the covid period. In fact, the aim of the study is to investigate the side effects of the vaccine in people who have had the disease or not. However, distinguishing side effects for different vaccines could provide more information".
The Reviewer also wrote: "Lines 207- 208: Is the number of vaccines administered to these people less than other groups? It would be appropriate to clarify this issue. Evaluate according to the sentence below:
“During the follow-up, 5743 subjects deceased for any cause (1.82% of the population; Table 2). The infected subjects were younger on average than the uninfected (42.9y vs. 50.7y, respectively) and showed a lower death rate (1.11% vs. 2.18%; Table S1)".
Yes, due to the very different restriction policies that were applied according to infection status, the proportion of subjects who received three or more doses of vaccine was significantly lower in the infected population (44.0%) than among the uninfected subjects (69.6%). This definitely created a bias that was apparent in some of the results. Please acknowledge that we tried to better clarify this point adding the following paragraph in the Introduction section: "In Italy, during the vaccination campaign, among the pandemic control policies, the following obligations were set to access a number of environments (including universities, hospitals, sport facilities and any public places), and for all the citizens aged 50 years or more: one COVID-19 vaccine dose for the subjects who were previously infected with SARS-CoV-2, or three vaccine doses for the uninfected subjects [14]. As a result, two different scenarios emerged. On one side, the infected population opted to receive one, two of three doses according to their personal convincement. On the other side, among the uninfected, it is likely that the strong pressure to receive at least three doses introduced a selection bias, according to which the subjects who received three doses might be different from those who received only one or two doses, as the latter individuals were reasonably discouraged to further vaccinations because of the occurrence of a disease, or since they deceased before the third dose. In order to account for the potential impact of the above mentioned bias, we will stratify the analyses by doses of vaccine and infection status". Also, please acknowledge that, to rule out the potential bias due to SARS-CoV-2 infection status and age, multivariate analyses were adjusted for age and other potential confounders and were also stratified by infection status.
II-2. The Reviewer wrote: "Line 236: It would be appropriate to examine the cause of mortality in these cases; these are the underlying malignancy etc. Is it people who have a problem who can't develop enough protective antibodies?: “Conversely, the uninfected subjects who received only one or two doses showed a significantly higher mortality than the unvaccinated”
The Reviewer also wrote: "Line 243-245: Why did those who received two doses of the vaccine have more side effects than those who received three or more doses? Is there a situation like immune blinding here?:
“A similar overall incidence of PVR-SAE was observed among the infected, while in the uninfected group the subjects who received only one or two doses showed a substantially higher proportion of events than those who received three or more doses“".
We absolutely agree that this point is very complex and deserves discussion. Please acknowledge that we added a paragraph in the Introduction section, and wrote the following paragraph in the Discussion section: "As mentioned, while the results were homogeneous for the infected population, the incidence of several outcomes, including death and total PVR-SAEs, was substantially higher among the uninfected subjects who received one or two vaccine doses, as compared to those who received three of more doses. This counterintuitive finding may be explained, at least in part, by selection bias. Indeed, during the pandemic, millions of Italians were obliged to receive three vaccine doses to retain their job or access a number of environments (including universities, hospitals, and public places) [14]. However, the subjects who were infected with SARS-CoV-2 were granted a six-month permission, after which they were mandated to receive only one vaccine dose. As a result, two different scenarios emerged. On one side, among the uninfected subjects, given the strong pressure to receive three doses, there were only two reasonable categories: those who refused the vaccine in toto, and those who wanted to receive three doses, as receiving one or two doses only would have not solved the mandate issue. This is indeed apparent in the sample: 83.2% of the vaccinated uninfected subjects received at least three doses. It may thus be assumed that many of those who received only one or two doses were discouraged to further vaccinations because of the occurrence of a disease, or deceased before the third dose. Thus, these subjects were selectively retained into the groups "one dose only" and "two doses only". On the contrary, the infected population did not experience the above selection bias: the vaccination mandate was far less stringent, and individuals opted to receive one, two of three doses according to their personal convincement, which indeed resulted in very homogeneous results across all vaccine groups. In any case, and most importantly, even among the uninfected individuals, the overall comparison, which included all subjects who received at least one dose, showed no increase of any outcome for the vaccinated subjects, as compared with the unvaccinated".
II-3. The Reviewer wrote: "Line 251-252: Please comment on the following sentence in the discussion section: “Compared with the unvaccinated, the subjects who received at least one vaccine dose showed a significantly lower likelihood of PVR-SAE (adjusted HR: 0.39; 95% CI:0.36-0.43; Table 3)”".
We agree and accordingly wrote the following sentences in the Discussion section: "In the entire population of an Italian province, followed for an average of 14 months, the subjects who received one or more doses of COVID-19 vaccines did not show an increased risk of death for any cause, death unrelated to SARS-CoV-2 infection, or any of the recorded potentially vaccine-related serious adverse events, requiring hospitalization, with some of the outcomes (all-cause mortality, and incidence of stroke, cardiac arrest, thrombosis, and pulmonary embolism) significantly less frequent among the vaccinated. These findings were consistent across genders, age-classes, most frequently administered vaccine types, and SARS-CoV-2 infection status, and remained after the adjustment for previous episodes of disease and several potential confounders. While the effectiveness of these vaccines against severe COVID-19 was documented in several population studies [2-6], some primary studies [7-,9-11,23] and secondary analyses of serious adverse events reported (a) in the randomized clinical trials of Pfizer and Moderna mRNA COVID-19 vaccines in adults [16,24], (b) in the U.S.A. Vaccine Adverse Event Reporting System (VAERS) passive surveillance data [25,26], (c) in the National registries of Denmark, Finland and Norway [27], raised concerns about the safety or risk-benefit profile of COVID-19 vaccination, especially in the subsets of population with the lowest infection-fatality rate [28]. However, other studies did not show a higher risk of serious adverse events in mRNA vaccine recipients [29-31]. Most importantly, the only two population-based primary studies with a control group available to date found a significantly lower non-COVID mortality among the vaccinated individuals after 28 days [12] and 7.5 months of follow-up [8]".
Reviewer 3 Report
This study is an attempt to use data from the entire population (> 6 yr.) of an Italian province (Pescara) to determine the [adverse] effects of COVID-19 vaccination. Database covered period from Jan 2021 to July 2022 (eighteen months), and if the individuals succumbed to COVID-19 infection, adverse reactions were only considered before the infection. The authors stratified vaccination into number of doses [vaccination] into more than one, one, two, three, and more than three doses. They found that there were no increased all-courses deaths, non-COVID-19 deaths, nor any potentially vaccine-related serious adverse events (PVR-SAEs) across genders and age-groups, with or without subsequent COVID19 infection. However, those not subsequently infected with less than three doses [completed vaccination] vaccinees had more PVR-SAEs. The authors attributed that might be due to “sampling bias”, as these individuals were “restricted” by public health policies.
The scale of this study is big regarding the entire (adult) population [n=316,315] and with an established healthcare network and follow-ups. Useful information and potentially validated new knowledges may be generated from this scale of study. However, this manuscript has numerous flaws and deficiencies:
1. Conclusions made by analysis of grouping together sometimes contradicted from the stratified data (Supplementary tables). For examples, in S1, it is clearly that age >60 yr. with 2-doses had a higher mortality. Likewise, this age group and doses had higher other PVR-SAEs than any other groups.
2. It is confusing for the groups uninfected vs. unvaccinated.
3. The rationale for stratifying into multiple vaccination doses is unclear. Current literature indicates that the second dose with mRNA COVID-19 vaccine results in higher adverse reactions.
4. Whereas the subsequently infected vaccinees were excluded from the analysis, but in Table 3, it shows that the uninfected had a higher deaths and PVR-SAEs than “infected only”. Is it due to the fact that the “subsequent infected” had a shorter follow-up period, hence less PVR-SAEs?
5. Only death and cardiovascular events were included. It is not clear why other adverse effects were not analyzed (except in Table 1, diabetes, COPD, kidney disease and cancer)
6. Table 2 should be divided into Table 2A and Table 2B
7. There were too many endnotes for the Tables (1-3), they should be summarized in the Materials and Methods
8. It is not clear what Fig. 2 supposed to show. For unvaccinated and the single-dose recipients, are age and COVID-19 infections play a role in these deaths?
In summary, this manuscript is potentially interesting, but it needs a major revision by:
1. Instead of stratifying into doses, it should be stratified into partial and complete vaccination and/or the duration of “follow-up” for these PVR-SAEs.
2. Justification and validation on using Kaplan-Meier Survival Estimates for Fig. 1
3. Elaborate on Fig. 2.
4. The abstract is confusing in regard to uninfected and infected. Needs to clarify
Author Response
III-1. The Reviewer wrote: "This study is an attempt to use data from the entire population (> 6 yr.) of an Italian province (Pescara) to determine the [adverse] effects of COVID-19 vaccination. Database covered period from Jan 2021 to July 2022 (eighteen months), and if the individuals succumbed to COVID-19 infection, adverse reactions were only considered before the infection. The authors stratified vaccination into number of doses [vaccination] into more than one, one, two, three, and more than three doses. They found that there were no increased all-courses deaths, non-COVID-19 deaths, nor any potentially vaccine-related serious adverse events (PVR-SAEs) across genders and age-groups, with or without subsequent COVID19 infection. However, those not subsequently infected with less than three doses [completed vaccination] vaccinees had more PVR-SAEs. The authors attributed that might be due to “sampling bias”, as these individuals were “restricted” by public health policies. The scale of this study is big regarding the entire (adult) population [n=316,315] and with an established healthcare network and follow-ups. Useful information and potentially validated new knowledge may be generated from this scale of study. However, this manuscript has numerous flaws and deficiencies".
The Reviewer also wrote: "Conclusions made by analysis of grouping together sometimes contradicted from the stratified data (Supplementary tables). For examples, in S1, it is clearly that age >60 yr. with 2-doses had a higher mortality. Likewise, this age group and doses had higher other PVR-SAEs than any other groups".
We agree and thank the Reviewer for his/her comments. It is absolutely true that the elderly subjects who received 2 doses showed a higher mortality than the unvaccinated. However, please acknowledge that this result are in line with the overall results, which are derived from multivariable analyses. Indeed, as shown in Table 3, in the overall sample the subjects who received two doses only showed a significantly higher likelihood of both death and overall PVR-SAEs.
III-2. The Reviewer wrote: "It is confusing for the groups uninfected vs. unvaccinated". The Reviewer also wrote: "The rationale for stratifying into multiple vaccination doses is unclear. Current literature indicates that the second dose with mRNA COVID-19 vaccine results in higher adverse reactions". The Reviewer also wrote " Instead of stratifying into doses, it should be stratified into partial and complete vaccination and/or the duration of “follow-up” for these PVR-SAEs". The Reviewer finally wrote: "Justification and validation on using Kaplan-Meier Survival Estimates for Fig. 1".
We agree that we should have clarified these points much better, and we accordingly added the following paragraph in the Introduction section: "In Italy, during the vaccination campaign, among the pandemic control policies, the following obligations were set to access a number of environments (including universities, hospitals, sport facilities and any public places), and for all the citizens aged 50 years or more: one COVID-19 vaccine dose for the subjects who were previously infected with SARS-CoV-2, or three vaccine doses for the uninfected subjects [14]. As a result, two different scenarios emerged. On one side, the infected population opted to receive one, two of three doses according to their personal convincement. On the other side, among the uninfected, it is likely that the strong pressure to receive at least three doses introduced a selection bias, according to which the subjects who received three doses might be different from those who received only one or two doses, as the latter individuals were reasonably discouraged to further vaccinations because of the occurrence of a disease, or since they deceased before the third dose. In order to account for the potential impact of the above mentioned bias, we will stratify the analyses by doses of vaccine and infection status".
As regards the duration of follow-up, please acknowledge that we account for the potential effect of follow-up duration using a survival analysis (Cox proportional hazards model), which is the standard method in longitudinal analyses with variable follow-ups. Similarly, please acknowledge that Kaplan-Meier survival curves are the gold-standard for this type of analyses.
III-3. The Reviewer wrote: "Whereas the subsequently infected vaccinees were excluded from the analysis, but in Table 3, it shows that the uninfected had a higher deaths and PVR-SAEs than “infected only”. Is it due to the fact that the “subsequent infected” had a shorter follow-up period, hence less PVR-SAEs?"
We agree that this point is confusing and should have been better clarified, and we are sorry for that. We replaced the previous, confusing sentence in the Methods "For the subjects who had a PVR-SAE, only the infections that occurred before the outcome were considered" with "For the subjects who had a PVR-SAE, only the subjects who had a positive swab before the outcome were considered as infected; those who were infected after the outcome were considered as uninfected for that specific analysis". Please acknowledge that the subsequently infected subjects were not excluded from the analysis.
III-4. The Reviewer wrote: "Only death and cardiovascular events were included. It is not clear why other adverse effects were not analyzed (except in Table 1, diabetes, COPD, kidney disease and cancer)".
We agree that this important point should have been better clarified, and we accordingly replaced the previous sentence in the Methods section "We selected 14 outcomes based upon frequency and clinical severity from a priority list of potential adverse events of special interest" with "We selected 14 outcomes based upon frequency and clinical severity from a priority list of potential adverse events of special interest produced by the Brighton Collaboration and Coalition for Epidemic Preparedness, Innovation Partnership, Safety Platform for Emergency Vaccines, in their secondary analysis of serious adverse events reported in phase 3 randomized clinical trials of Pfizer and Moderna mRNA COVID-19 vaccines in adults [16]. Please acknowledge that this list was endorsed by the WHO Globally Advisory Committee on Vaccine Safety and, unfortunately, other important possible outcomes - such as sudden death - could not be evaluated with our data sources".
III-5. The Reviewer wrote: "Table 2 should be divided into Table 2A and Table 2B".
We agree and accordingly divided Table 2 into Table 2A (showing the results of the outcome "death") and Table 2B (showing the results of the other potentially vaccine-related serious adverse events).
III-6. The Reviewer wrote: "There were too many endnotes for the Tables (1-3), they should be summarized in the Materials and Methods".
We entirely agree and accordingly cut the details on the ICD-9-CM codes and comorbidities extraction codes from Tables footnotes, as they were already reported in the text. Please acknowledge that we also synthesize the groups definitions. We thank the Referee for the suggestion.
III-7. The Reviewer wrote: "It is not clear what Fig. 2 supposed to show. For unvaccinated and the single-dose recipients, are age and COVID-19 infections play a role in these deaths?" The Reviewer also wrote: "Elaborate on Fig. 2".
We agree that Figure 2 deserves a clarification, and we apologize for not having provided it before. As shown in Table S1 (and as reported in the Results section, lines 226-233) the overall mortality was similar among the subjects with a previous infection who received either one, two or three (or more) vaccine doses. Conversely, among the uninfected, the mortality was significantly lower among those who received three or more doses (0.70%) as compared to those vaccinated with two (6.65%) or one dose only (5.70%). This scenario may be attributed to the above mentioned selection bias, according to which, given the strong pressure towards the uninfected to receive at least a complete immunization schedule (three doses), the subjects who were immunized with three doses might be different from those who received only one or two doses, as the latter individuals were reasonably discouraged to further vaccinations because of the occurrence of a disease, or since they deceased early, in any case before they could receive further doses. This scenario seem to be confirmed by the data reported in Figure 2, showing a substantially higher proportion of early deaths among the subjects who received a single dose (63.3% of the deaths occurred within the first 60 days of follow-up), while, among the unvaccinated, only 30.8% of the deaths occurred within the first 60 days of follow-up.
Please also acknowledge that, to rule out the potential bias due to SARS-CoV-2 infection status and age, multivariate analyses were adjusted for age and other potential confounders and were also stratified by infection status. As reported in Table 3, the same scenario above described emerged in the multivariate analyses: among the infected individuals, there was a comparably lower likelihood of death in those vaccinated with one, two or three or more doses versus the unvaccinated. Conversely, among the uninfected individuals, there was a significantly lower likelihood of death with three or more doses (versus none), but a significantly higher likelihood of death after one or two doses, versus zero.
III-8. The Reviewer wrote: "The abstract is confusing in regard to uninfected and infected. Needs to clarify".
We agree and accordingly replaced the previous sentence in the Abstract "In the infected population, the results did not vary by vaccine dose", with "In the infected population, every dose of vaccine was associated with a lower risk of death and PVR-SAE".
Round 2
Reviewer 3 Report
The authors only made superficial changes, or revision, to this manuscript. The significant points raised in my previous review report were not addressed.
After reading the authors’ reply, it raised more questions.
The authors want to point out that with a minimum of ONE dose, the fatality rate is reduced. But according to the dichotomy below, the stratification should be:
A. Uninfected cohorts
a. Uninfected throughout
i. 1 dose
ii. 2 doses
iii. ≥3 doses
b. Infected [after immunization]:
i. 1 dose
ii. 2 doses
iii. ≥3 doses
B. Infected cohorts [before immunization]:
c. Uninfected throughout?
i. 1 dose [=just one dose?]
ii. 2 doses
iii. ≥3 doses
d. Infected:
i. 1 dose
ii. 2 doses
iii. ≥3 doses
The grouping scheme is confusing to this reviewer, and presumably will be more confusing to the readers. Whereas it is understood that with prior infection, some degree of immunity is elicited, hence requiring just one dose. However, grouping of this cohort to the [originally] uninfected cohort (with one dose) is problematic. Either the authors should trim off a subset of the data to conduct a proper analysis, or the authors should clarify these stratification and grouping scheme from the beginning, with the numbers clearly identified.
There are problems for Table 1 and Table 2 as well. The column ">1 dose" should be placed either as the last column (since it is the sum total for the columns "1 dose", "2 doses" and ">3 doses"), or at a level higher than these three columns. These tables, in their current format, are showing "double counting" error.
Line 80 to Line 108 should be summarized in a Table format, e.g., as Table 1 (and the subsequent tables renumbered accordingly). Also, the footnotes for the Tables should be in Roman Numbers, e.g., I, II, III, etc., rather than in Capital letters.
Author Response
III-1. The Reviewer wrote: "The authors only made superficial changes, or revision, to this manuscript. The significant points raised in my previous review report were not addressed. After reading the authors’ reply, it raised more questions. The authors want to point out that with a minimum of ONE dose, the fatality rate is reduced. But according to the dichotomy below, the stratification should be:
- Uninfected cohorts
- Uninfected throughout
- 1 dose
ii.2 doses
iii. ≥3 doses
- Infected [after immunization]:
i 1 dose
- 2 doses
iii. ≥3 doses
- Infected cohorts [before immunization]:
- Uninfected throughout?
- 1 dose [=just one dose?]
- 2 doses
iii. ≥3 doses
- Infected:
- 1 dose
- 2 doses
iii. ≥3 doses
The grouping scheme is confusing to this reviewer, and presumably will be more confusing to the readers. Whereas it is understood that with prior infection, some degree of immunity is elicited, hence requiring just one dose. However, grouping of this cohort to the [originally] uninfected cohort (with one dose) is problematic. Either the authors should trim off a subset of the data to conduct a proper analysis, or the authors should clarify these stratification and grouping scheme from the beginning, with the numbers clearly identified".
We entirely agree. Please accept our apologies for not having understood the previous revision. Please acknowledge that we further stratified the infected subjects as requested, splitting them into "infected before vaccination" and "uninfected before vaccination". We thus re-run all analyses on the main outcomes comparing the following groups:
- Unvaccinated, uninfected throughout
- Unvaccinated, infected during the follow-up
- 1 dose only, infected before vaccination
- 1 dose only, infected after vaccination
- 2 doses only, infected before vaccination
- 2 doses only, infected after vaccination
- 3 or more doses, infected before vaccination
- 3 or more doses, infected after vaccination
- At least 1 dose (including those who received 1, 2, 3, or 4 doses), infected before vaccination
- At least 1 dose (including those who received 1, 2, 3, or 4 doses), infected after vaccination
We reported all univariate and multivariate results in a newly created Table (S15). Overall, no substantial differences were found between those who were infected before or after the vaccine. We discussed the new analyses in the Discussion section, adding the following paragraph: "Both epidemiological and immunological studies showed that COVID-19 vaccination, received either before or after SARS-CoV-2 infection (known as "hybrid immunity"), is able to increase the protection against the virus as compared with the natural immunity [15,32]. No data are available, however, on the potential impact on vaccination safety of having received the vaccine prior or after vaccination. We thus further stratified the infected population according to the timing of the infection (Table S15), finding no substantial differences in the overall results between the individuals who were infected before or after vaccination".
III-2. The Reviewer wrote: "There are problems for Table 1 and Table 2 as well. The column ">1 dose" should be placed either as the last column (since it is the sum total for the columns "1 dose", "2 doses" and ">3 doses"), or at a level higher than these three columns. These tables, in their current format, are showing "double counting" error".
We entirely agree, and thank the Reviewer for his/her suggestion. Please acknowledge that the data referred to the "≥1 dose" group were moved to the last column in all tables. Please also acknowledge that all the footnotes were re-numbered accordingly.
III-3. The Reviewer wrote: "Line 80 to Line 108 should be summarized in a Table format, e.g., as Table 1 (and the subsequent tables renumbered accordingly). Also, the footnotes for the Tables should be in Roman Numbers, e.g., I, II, III, etc., rather than in Capital letters".
We entirely agree that these methodological details could have been made clearer. Accordingly, we summarized the follow-up time-windows for each group of subjects according to vaccination status into a new Figure, titled "Figure 1. Flowchart of study participants", and all the other figures were re-numbered accordingly. Also, we re-numbered all Tables footnotes using Roman numbers.
Round 3
Reviewer 3 Report
This is a significantly improved revision. Inclusion of a flowchart (Fig. 1) is a plus.